# Engaging Diverse Community Groups to Promote Population Health through Healthy City Approach: Analysis of Successful Cases in Western Pacific Region

**DOI:** 10.3390/ijerph18126617

**Published:** 2021-06-19

**Authors:** Albert Lee, Keiko Nakamura

**Affiliations:** 1Centre for Health Education and Health Promotion, The Chinese University of Hong Kong, Shatin, Hong Kong, China; 2Secretariat, Alliance for Healthy Cities, Tokyo 101-0062, Japan; nakamura.ith@tmd.ac.jp; 3Department of Rehabilitation Science, Hong Kong Polytechnic University, Hong Kong, China; 4WHO Collaborating Centre for Healthy Cities and Urban Policy Research, Graduate School, Tokyo Medical and Dental University, Tokyo 113-8519, Japan

**Keywords:** health promotion, community engagement, healthy city, healthy setting, determinants of health, built environment, public health, urban governance, SPRIRIT framework, Alliance for Healthy Cities

## Abstract

Background: A substantial global burden of health can be attributed to unhealthy lifestyles and an unhealthy living environment. The concept of a Healthy City is continually creating and improving physical and social environments to enable healthy living. The aim of this paper is to investigate how the Healthy City concept would tackle the complexity of health by addressing the socio-economic and political determinants of health in the Western Pacific Region. Methods: The SPIRIT model adopted by the Alliance for Healthy Cities can provide a framework for an integrated and holistic approach to enable policy, environment, social matters, behaviours, and bio-medical interventions to take their rightful place side by side. The performance of cities awarded by the AFHC was analysed under each domain of the SPIRIT model to show the efforts striving to acquire the qualities of a healthy city. Findings: Two cities have incorporated the Healthy City concept in most of their policies outside the health sector, with a high level of commitment from city leaders and citizens, so the Health City activities were recognised as part of the means to advance the cityies’ general planning. One city has made use of its strong network of key stakeholders from different sectors and disciplines to establish a “Medical–Social–Community’ model. All three cities have collected health information to reflect health status, determinants of health and issues reflecting health promotion to enable the creation of a city health profile and show positive changes in health. The cities have engaged key stakeholders to launch a variety of health-promoting programmes according to the needs of the population. Conclusion: The AFHC can play an important role in linking the cities with strong action in Healthy City activities to support other cities in Healthy City development.

## 1. Background and Introduction

### 1.1. Environment as a Key Determinant of Health

Behavioural, environmental and occupational, and metabolic risks can explain half of global mortality and more than one-third of global Disability-Adjusted Life Years (DALYs), providing many opportunities for prevention [1]. A substantial burden of global cardiovascular disease (CVD) morbidity and mortality is attributable to a sedentary lifestyle, and the attributable burden of a high BMI has increased over the past 23 years, and physical inactivity and unhealthy eating are the underlying cause [2]. Non-communicable diseases (NCDs) such as cardiovascular, chronic lung diseases, cancer, and diabetes are constituting major health burdens, and the main drivers for an unhealthy diet and lack of physical activities are due to lack of places and opportunities to be physically active as well as industry opposition to public health interventions [1,2,3]. Physical inactivity has been shown to link with poor outcomes for physical and mental health, leading to chronic illnesses and increased mortality [4]. Socio-ecological models state that a person’s health status is not only influenced by individual behaviour but also by factors situated in the person’s environment, so the improvement of health requires modification of behavioural and environmental risks to be more conducive for healthy living [5]. The new public health described by Ashton and Thurston has highlighted the city as a place shaping human possibility as an ecological concept of health, and it has a crucial role to play in determining the health of those living in it [6]. The paper has also recapped the philosophy of the two founders of Healthy City (HC), Hancock and Duhl, “A healthy city is one that is continually creating and improving those physical and social environments and expanding those community resources which enable people to support each other in performing all the functions of life and developing themselves in their maximum potential” [7]. Therefore, there is a need to analyse how cities would make use of the HC concept as a socio-ecological model to improve health. This paper is to report the case studies of three successful Healthy Cities in the Western Pacific Region.

### 1.2. Impact of Built Environments on Physical and Mental Health

The Built Environment (BE) can be defined broadly as “the human-made space in which people live, work and recreate on a day-to-day basis” [8]. It relates to objective or perceived environment, variation in spatial scale, and how its boundaries are drawn. There is an association between the BE and obesity mediated by physical inactivity and sedentary behaviour of using cars; changes in the BE, such as increased land mixed use (residential, commercial, office, and institutional), residential density, and street connectivity making walking an attractive and viable option are predicting factors for obesity [9]. The BE is not only green spaces and parks but also the conditions of sidewalks, traffic flow, cleanliness and maintenance of public spaces, perceptions of safety and community security, zoning and land use mix, and population density [10]. The BE also includes the internal environment and social capital (defined as social networks and interactions that inspire trust and reciprocity among citizens) [11]. Better street connectivity/walkability tends to be positively related to physical activity and walking, and urbanicity, measured by population density, is associated with a higher probability of smoking and alcohol consumption (except binge drinking) [12]. BE factors such as walkability [13] and green space [14] affect diverse health outcomes, health behaviours [15], and risk factors [16]. Neighbourhoods characterised as more walkable, either leisure-orientated or destinated-driven, are associated with increased physical activity, increased social capital, a lower overweight population, lower reports of depression and less reported alcohol use as reported by a systematic review [17]. Findings from a study in Seoul, Republic of Korea (Korea) has indicated the need to consider various design techniques to encourage people to walk for leisure and opportunities for networking [18]. Findings of a study conducted in South Africa have highlighted the importance of green space as protective against the onset of depression [19].

### 1.3. Impact of COVID-19 on Physical and Mental Health

The world was very much affected by COVID-19 infection during the year 2020. The social distancing, self-isolation, and “lock down” measures have made the BE promote physical inactivity [20]. A study conducted in Hong Kong has shown that 65.3% of participants experienced increased stress due to staying at home and 29.7% experienced moderate to severe levels of depressive symptoms, and increase in the use of electronic devices and the decrease in outside activities was positively associated with a higher level of depressive severity [21].

Another study in Hong Kong has also shown a significant increase in risk and anxiety during the COVID-19 outbreak and compared with 2016 and 2017, the stress level increased by 28.3%, the prevalence of anxiety increased by 42.3%, and depression symptoms and unhappiness have doubled (all p for trends < 0.001) [22]. The increases in stress levels were significantly larger among older and less-educated respondents. 

COVID-19 also has an impact on physical health. A study has found that parents of older children (ages 9–13) compared with younger children (ages 5–8) perceived greater decreases in physical activity (PA) and greater increases in sedentary behaviours (SB) from the pre- to early COVID-19 periods, and children were more likely to perform PA at home indoors or on neighbourhood streets during the early vs. pre-COVID-19 periods [23]. There is a concern that these short-term changes in PA and SB in reaction to COVID-19 may become permanently entrenched, leading to increased risk of obesity, diabetes, and cardiovascular disease in children [23]. An Italian study has shown a significant decrease in total physical activity before and during the COVID-19 pandemic in all age groups and especially in men, and a significant positive correlation was found between the variation of physical activity and mental well-being, suggesting that the reduction of total physical activity had a profoundly negative impact on psychological health and well-being [24]. It becomes a top priority to formulate strategies to combat the adverse impact of COVID-19 on population health and prevent a “sick city” from further isolating and segregating the difference, and drawing a collective strength from its mixture of different people based on the philosophy of HC [6,7].

### 1.4. Healthy City Concept to Create a Healthy Supportive Environment

Phase VI of the European Healthy City supports the cities in strengthening their efforts to bring key stakeholders together to work for health and well-being, harness leadership, innovation, and change, and enhance the potential to resolve local public health challenges [25]. A healthy city is continually creating and improving physical and social environments and expanding community resources which enable people to support each other in performing all the functions of life and maximise their potentials in the socio-ecological model of health improvement [6,7]. A study in Japan has also found the effect of a quality living environment on behavioural outcomes in children after controlling for socio-economic factors [26]. The study has found that greater aesthetic quality, walkability, accessibility of healthy foods, safety and social cohesion were inversely linked to children’s behavioural problems and positively linked to social competence [26]. Neighbourhoods characterised as more walkable (leisure or destination-driven) are associated with increased social capital [17,18]. Perceptions of higher social capital correlate with better mental health [27]. 

An important relationship exists between the BE (physical and social) and health (physical and mental). It would add value to draw on successful examples of Healthy Cities to promote a healthy living environment in a broader perspective covering both physical and social perspective as stated in the 2016 Kuching Statement that “the hallmark of successful 21st century cities will be an understanding of urban development in terms of the complex interconnections between the ecological, economic and social foundations of human development and health” [28]. COVID-19 would turn the priority of the health agenda to focus on a medical model, particularly the bio-medical perspective, ignoring the social model of health. The aim of this paper is to conduct case studies among cities in the Western Pacific Region that have been awarded by the Alliance for Healthy Cities (AFHC) for their achievements in the complex interconnections to promulgate HC development.

## 2. Materials and Methods

### 2.1. Conceptual Framework of Accreditation of Healthy Cities by AFHC

Any city can start the process of becoming a HC if it is committed to the development and maintenance of physical and social environments by supporting and promoting better health and quality of life for residents and building health considerations into urban development [6,7]. A HC involves incorporating and/or accounting for multiple and complex factors, and there is a need for action across the whole of society focusing on those social factors that determine health outcomes [29], so management is crucial for Healthy Cities. HC networks have been expanded to all six WHO regions, including the Western Pacific Region, with many emerging megacities. While the WHO has expertise in the HC approach, it is the city that actually makes a HC plan and carries out the HC programme. International cooperation is an effective and efficient tool to achieve the ambitious goal. The Alliance for Healthy Cities (AFHC) was established following a WHO consultation meeting in 2003, and the membership cities have expanded from 10 to over 170 at the time of writing [30]. AFHC is an international network aiming at protecting and enhancing the health of city dwellers by promoting the exchange of people who are in the front line of health issues across different sectors and disciplines [30]. 

The AFHC advocates the key features of the HC project, including high political commitment, intersectoral collaboration, community participation, integration of activities in elemental settings, development of a city health profile and a local action plan, periodic monitoring and evaluation, participatory research and analyses, information sharing, involvement of the media, incorporation of views from all groups within the community, mechanisms for sustainability, linkage with community development and human development, and national and international networking [30]. Taking into account the diverse needs for a Healthy City and the appropriate evaluation tools for health promotion and a healthy setting, the SPIRIT framework was proposed to the AFHC to evaluate cities for an award and was adopted by the AFHC General Assembly in October 2004 and subsequent Mayors’ meeting and AFHC Global Conference [31,32]. Apart from taking a purely scientific view, evidence is needed to guide the actions of intervention developers and decision-makers. The SPIRIT framework addresses Setting and Sustainability, Political commitment and Policy, Information and Innovation, Research and Resources, Infrastructure and Intersectoral, and Training. The model can provide a framework to pursue an integrated and holistic approach to public health so that policy, environment, social matters, behaviours, and bio-medical interventions can take their rightful place side-by-side [6,33,34,35,36,37]. Positive outcomes from those components would reflect the success of a HC, and the model can enable the application of the HC concept to be evaluated and generate good case studies for analysis [38]. The framework is not measuring whether the city has achieved a particular level of health but is conscious of health and striving to improve it. It has taken into account the diversity of cities, such as size and scale of city development. One should plan our cities as Healthy Cities with key strategies to enable us to live well and healthy as outlined in Table 1. We can develop case studies to learn the process of the cities to enable people to take control over and improving their health by applying the SPIRIT framework to evaluate their cities’ achievements.

Appendix A outlines the requirements under each domain;the key issues of the domains are:

Setting approach, Sustainability

The settings approach can be an effective way to create supportive environments, as it can facilitate complex interventions to suit particular settings, mechanisms to secure political commitment, healthy public policy, intersectoral collaboration, community participation, information sharing, and resources important to assure sustainability. The concept of a super setting approach can optimise the use of diverse and valuable resources embedded in the local community and strengthen social interaction and local ownership as drivers of the changing process [39]. 

Political commitment, Policy, Partnership

Healthy public policies, when effective, can be a threat to some established interests and existing powerful group, so the balance should not be only based on political goodwill and commitment of the local politicians and administrators, but also the community [33]. Equity in health and human-centred sustainable development are core values for the new European Health Policy Framework and Strategy, and cities having significant influences on the wider determinant of health and political commitment is one of the important actions [35].

Information, Innovation

Quantitative and qualitative information in assessing the determinants of health should be utilised to include demography, household, education, income and family expenses, employment and occupation, local economy and industry, infrastructure, living environment and sanitation, housing environment, environmental quality, land use, urbanisation, community activities, lifestyle and health behaviours, disease-prevention activities, health care and welfare services, cultural values, and leisure and recreational services [40,41]. Programmes should be innovative in meeting public needs and promote a climate to support changes. 

Resources, Research

A basis of evidence is essential to develop HC for policy making, rational decision making, effective planning, efficient allocation of resources, visible evaluation of outcomes, and fruitful exchange of experiences with other cities [42]. Healthy City cross-cutting health initiatives in cities needs to address the needs for the marginal population to justify higher-end development so research guiding the allocation of resources is essential [43]. 

Infrastructure, Intersectoral

An urban health systems model focusing on multisectoral approaches looking beyond the health sector can tackle the determinants of health and recognise the plurality of health service providers [44], as it is difficult for any single institutions to restore all issues [37].

Training

One must observe the spirit of capacity building in the community, so training is essential for the successful implementation of HC.

### 2.2. SPIRIT Framework to Identify Successful Experiences of Healthy Cities

AFHC has launched the AFHC Awards, not only as an accreditation of good practice of healthy cities but also as a way to foster the sharing of good practice. The criteria for award accreditation are based on the SPIRIT framework [38]. The three levels of AFHC Awards (Appendix A) synchronise nicely with the evaluation of health promotion programmes in terms of infrastructure/input, process, impact, and outcomes. The level 1 award emphasises building good infrastructure for HC development. Level 2 reflects the dynamic of the communities to create a supportive environment for health and the process of change. Level 3 reflects the impact of policies and actions beyond the health sector with positive changes in the health of the population (outcomes). The three case studies have reached level 3, so they can demonstrate the evolutionary changes.

Korea has developed a nationwide HC movement, and the Government has launched an evidence-based HC programme by conducting a Health Impact Assessment [45,46]. The health-promotion capacity map in Korea has made a good initiation to get started and what is needed next is a more “fine-tuned” map acting as “radar traps” to show “speed bumps”, “construction sites”, and “one-way streets”. Korea had made use of the SPIRIT framework to identify city problems and sought a way for its improvement based on the HC project philosophy and strategies by engaging key stakeholders as well as acquiring the qualities of Healthy Cities [47]. An analysis of the performance of cities with an AFHC award would identify some of the key factors for success.

### 2.3. Case Studies of Cities with an AFHC Award for a Healthy City with Strong Action

The important consideration of the success of HC is not whether the city has achieved a particular level of health; it is whether the city is conscious of ways to improve the health of residents through strong political commitment, community engagement, and strong collaboration with different key stakeholders [38,48,49,50,51]. The HC concept is about a process rather than just outcomes, so the evaluation methodology needs to enable the cities to act upon the needs of the community by making changes to policies, services and organisational structure and collaborative processes. The cities, Gangdong-gu (Gg) in Korea, Owariasahi City (OC) in Japan, and Kwai Tsing District (K and T) in the Hong Kong Special Administrative Region (HKSAR) achieved an AFHC Award for a Healthy City with Strong Action in 2018. Appendix A provides a brief background of the three cities. The cities have put health in all policies with research studies on HC development and also engaged of stakeholders across different sectors with an administrative structure in place. These cities have demonstrated a long-term track record in HC movement with great success. OC was awarded Healthy City with Strong Action in 2010, 2012, and 2016 (Outstanding Performance), and Creative Development on NCD prevention in 2012. Gg was awarded Creative Development for a good health system an award for Good Infrastructure in 2012, and Outstanding Performance of a Healthy City in 2016. K and T received the Healthy City Pioneer Award for integrating different healthy settings to evolve the “Medical–Social–Community” model in 2010. 

The performance of those cities under each domain of SPIRIT model was analysed to show the efforts striving to become a HC engaging key stakeholders as well as acquiring the qualities of HCs. The findings would be useful for those cities striving to become Healthy Cities.

## 3. Findings

The findings will present the key achievement of the above-mentioned cities under the different domains of the SPIRIT framework.

### 3.1. Setting Approach, Sustainability

Gg and OC have developed long-term strategic plans for city development, incorporating HC in most of the policies of the cities outside the health sector, with a high level of commitment from city leaderships as well as citizens (Table 2). Gg formulated Korea’s first customised Healthy City Planning Guidelines, and in OC, the HC activities were recognised as part of the means to advance the City’s general planning. Both Gg and OC have put HC high on the political agenda to enable people in different settings to take control of and improve their health. Gg has committed resources to support healthy school environment (physical and psycho-social), created land space to promote physical activities, developed eco-friendly housing complex environment, and a Health Management Centre to provide one-stop services from prevention to diagnosis. The Gg plan for restoring city forests had 100,000 trees planted over 4 years until 2014 and created a Green-way Forest trail 2.7 km as a theme forest. The city had 300 civic groups engaged to restore the damaged forests. The city also restored the ecological stream Godeokcheon, a tributary stream of the Han River, which is 3.54 km in length and has been known as a Crab Stream. It has become the city centre, with facilities such as waterfront stage, floor fountain, athletic park, walkway, and a bicycle road, and they are blended with walking events.

OC has developed an eco-garden city full of greenery and conservation of forests as well as ecological ISO for homes. The renovation of the whole of OC involved land adjustment projects with roads, parks, and other urban infrastructure was maintained and updated. Gg and OC have put in strong efforts to create BE conducive for healthy living. 

K and T, through the Kwai Tsing Safe City and Healthy City Association (KTSCHCA) established in the late 1990s with core members of the District Council passionate about HC, together with key stakeholders from the community, government departments, providers of health, education and social services, academics, and professionals, has made use of this strong network from different sectors and disciplines to develop health promotion in diverse settings to establish a “Medical–Social–Community” model to improve the health of the local population, making use of government-earmarked funding for a Signature Project Scheme (SPS) (Table 2) [52]. Hong Kong is highly urbanised with a high population density, and K and T has made a strong effort to ensure safety in homes, housing estates, workplaces, and schools (Table 1). 

### 3.2. Political Commitment, Policy, Partnership

These three cities have actively engaged local citizens to generate opinions and are actively involved in translating into practice factors such as connectedness with cultural and biological heritage in Gg and ecological improvement in OC (Table 3). K and T has made use of a strong network through KTSCHCA to initiate a variety of screening services for the local population (Table 3). The “Medical–Social–Community” concept by K and T was then adopted in the Policy Address of HKSAR Government as a strategic direction to enhance community health and establish the first District Health Centre (DHC) in K and T operated by KTSCHCA (Table 2).

The pledge of the Mayor election (2018–2022) in Gg has incorporated the key components of HC with revision of relevant ordinances putting health in policies beyond the health sector such as housing, city planning, transport, environment (Table 3). The Gg Healthy City Master Plan created the “Active Living Environment” project implemented by the different divisions of the city government to benefit the whole population. Similarly, OC introduced the HC concept in most of the city’s policies (Table 3), and it created its own vision for a Healthy City based on the analysis of its own issues and conditions. Both Gg and OC have organsied large-scale participation of local citizens in developing projects to improve the living environment (Table 2).

### 3.3. Information, Innovation

The health information of Gg and OC have included indicators to reflect health status, determinants of health and issues reflecting health promotion to enable the creation of a city health profile, and innovations and new initiatives were based on the city health profiles (Table 4). Results from questionnaires from city development and the Second Healthy Asahi 21 Plan in OC have shown steady and certain improvement in citizens’ health awareness, and about 90% of citizens felt that they were healthy. K and T has collected research findings related to injury prevention (intentional and non-intentional) (Table 4). All three cities have engaged stakeholders from many different sectors, including private organisations, to launch a variety of health-promotion programmes according to the needs of the population.

### 3.4. Resources, Research

Gg and OC have developed a research framework for needs assessment and evaluation to monitor the performance of healthy city development with funding from a variety of government resources (Table 5). K and T, through KTSCHC, in collaboration with academics and professionals, has developed an injury-surveillance system (Table 5). All three cities have demonstrated improvement reflecting changing to more positive health and have received funding support for healthy city research and development (Table 5). The governments of Gg and OC have earmarked resources for the creation of city health profile and related research studies. In K and T, resources mainly came from resources of the District Council, SPS, and other sources of funding obtained by academic institutions, mainly ad hoc funding.

### 3.5. Infrastructure, Intersectoral, and Training

All three cities include key stakeholders, government officials, professionals and academics in the Steering Committee of HCP, and Gg has designated the Deputy Mayor to chair, and the HCP team is positioned in the Planning and Budget Division of the City office (Table 6). All three cities have actively engaged in international exchanges and workshops locally (Table 6). The public seminars/workshops were well attended, e.g., the public hearing for Healthy City legislation of Gg was attended by 300 people, and other forums were also well attended.

## 4. Discussion

The aim of this paper is to investigate how to implement the HC concept in a city to tackle the complexity of health by addressing the socio-economic and political determinants of health. HC is an ecological model for better health. It is not easy to sell this complex ecological model with the dominance of the medical model and will always hear the frustration of HC practitioners, “We say public health, they hear medicine; we say health they hear disease”, as highlighted by de Leeus and Simos [53]. This has underplayed the important contributions by other sectors/groups, such as education, social work, housing, transport, security, city planning, parks and recreation management, local citizens, and the private sector, which are needed to make the city healthy, not merely an absence of disease. Therefore, various dimensions reflecting how the city functions, such as social cohesion, democratic life, community capacity, and social capital, should be measured rather than routine health statistic such as mortality and morbidity [54]. The measurement of health status should also go beyond mortality and morbidity and address well-being and quality of life [55], and also the determinants of health [56]. The prevailing bio-medical paradigm has shown to be a key cultural barrier to translate Health in All principles in day-to-day practice [57]. This is the spirit of the SPIRIT framework to identify how the city can create better health for the local citizens in a real sense. The analysis from the case studies of the cities with an established track record in HC movement would enlighten how HC would create conditions for better governance, enhanced inter-sectoral action, and opportunities for Health in All Policies as well as greater resilience of programmes facing resource limitation and political turmoil [53]. 

The seven leading plans of OC were formulated to realise the three guidelines defined in OC HCP and became the policy framework of the city. With the strong political commitment, many structures have been in place with systematic collaboration between different government departments and sectors across many aspects of city life, resulting in renovation of the whole city to be more conducive for healthy living through the improvement of city infrastructure and facilities, leading to a better living environment (Table 2 and Table 3). The city has engaged the private sector apart from local citizens in taking actions in HCP, and also sought expert advice from different professionals and academics. The health information collected would identify health care needs to cater to the health conditions of special population groups such as the elderly through “Cycle for Good Health and Enhancing Health of the Elderly” as the leading city plan, Koala Class programme to cater to the developmental needs of young children, and Asapy Vaccination Navigation Schedule to manage immunisation programmes (Table 4). The city health profile has resulted in innovative high level of community engagement, such as the “General Plan Deliberative Council”, “Children’s Conference”, and “Children’s Canteen”. As HC movement has become an integral part of city planning, monitoring and evaluation have been built in to assess the Second Healthy Asahi 21 Plan, showing steady and certain improvement in citizens’ health awareness according to the effects from various efforts of the HC Programme, and about 90% of the citizen perceived themselves healthy according to the result of questionnaires (Table 4). 

A periodic assessment by questionnaire study was also conducted to assess whether targets set in the policies and key programmes in the Fifth General Plan have been achieved to be used as a basis in further city planning with good achievements, and positive results have been achieved (Table 5). About 53% of citizens replied that they knew about the city’s efforts to become a HC on city development in 2018 and doubled since 2006. The city has made proud of the achievements in the improvement of average healthy life expectancy and the average independent life during as well as maintaining a low percentage of elderly citizens requiring nursing care/cost care for the elderly. One can regard OC as an exemplary example of cities taking a comprehensive citywide approach to health promotion across different aspects of city life with success in integrating the HC concept into core city policy framework [58]. The city has made use of HC development as a quality-assurance cycle of continuous improvement using the SPIRIT framework as criteria and standard.

The development of the Master HC plan by Gg, being Korea’s first customised Healthy City Planning Guideline with the Mayor’s election pledge, has led to health in city policies across a range of sectors (Table 2 and Table 3). Gg has developed a very comprehensive city health profile with indicators reflecting HC development, including socio-economic determinants of health, and innovations and initiatives based on city health profile have covered healthy living from wide perspectives (Table 4). The research framework on needs assessment is also comprehensive, and the evaluation has shown positive changes (Table 5). Apart from the improvement of health behaviours as shown in Table 5, Gg’s health ranking according to the “K-Health Ranking” by Hallym University Institute of Social Medicine has raised from one-hundred-and-forty-seventh in 2008 to fourteenth in 2016, the highest growth rate in Korea. The Korean Society for Equity in Health has reported Gg has the smallest life expectancy gap between income levels, 3.8 years. Healthy cities are more than health intervention; they are value systems with rigorous preparation and in-depth consultation with all stakeholders, including communities and industries, and HC can then create conditions for better governance, enhancement of intersectoral action, and opportunities for Health in All Policies [59,60,61]. Both Gg and OC have established strong infrastructure for HC development, including city leaders, professionals, academic experts, and citizens. This would explain how HC development is high on the political agenda with a strong policy of commitment and community engagement and participation, with signs of improvement of the health of the cities’ populations with remarkable improvement in the living of their citizens. The tighter and broader partnership facilitating an enhanced system of governance would lead to more effective city-based programmes aiming to facilitate healthier lifestyle choices.

K and T have adopted a community development approach taking advantage of a strong network of professional and academic experts and the local community to develop health promotion across many settings. The healthy setting is a powerful ecological model to improve health as the focus is on interplay of range of environmental, social, behavioural and personal factors, an emergent model of causation, not a linear model [62]. There are two cities in Australia, Kiama/Illawara in New South Wales, and Noarlunga/Onkaparinga in South Australia that are the longest-running Healthy Cites in the world run by civil society foundations, which may or may not connect to an official government agenda. The political commitment to Healthy City Onkaparinga (HCO) has varied over the years, and the strong connection with Flinders University is the major contributor to the ongoing achievement [63]. The HCO has provided opportunities for students and scholars to see HC initiatives in action, and this has strengthened the academic focus with HCO. The multiple Heathy Settings approach in collaboration with leading academic institutions and professional bodies as a strategy for HC movement in K and T has led to wider coverage of the district population generating evidence of needs and effectiveness. The multiple Healthy Setting has enabled the Health-Promoting Hospital movement initiated by the main hospital to cascade the effects to other settings, linking up to create a safe and healthy community [64]. Building on years of experience in promoting safety and health in the community through the Healthy Setting approach, KTSCHCA established a “Medical-Welfare-Community Collaborative Model” to capitalise on the intersectoral partnership and multidi-sciplinary platform to build a sustainable, safe, and healthy community in K and T [52,64]. This has provided a platform for HKSAR Government to conduct a pilot scheme of DHC in K and T. K and T should continue maintaining a strong link with the academic and professional partners, so the HC movement in K and T has a strong academic focus to drive the development.

The three cities have built the capacity of individuals, families, and communities to create strong human and social capital through the HC movement. Social capital can also influence the health behaviours of neighbourhood residents by promoting a more rapid diffusion of health information, increasing the likelihood to adopt healthy norms of behaviour, and exerting social control over deviant health-related behaviour [65]. 

There are limitations of the case studies. The findings only reflect data being reported with no direct observation by a site visit. There were no opportunities to clarify and substantiate the information provided. There are no comparable studies of cities not performing well in HC movement to contrast the deficiencies under the SPIRIT framework. It is not feasible to have cities as the control group, as city development is an ongoing process responding to changing circumstances. It is neither practical nor ethical to put cities on hold. A more realistic approach should be in place for the evaluation of healthy cities [66].

Only Gg has presented some quantitative data longitudinally for monitoring and evaluation. The SPRIT framework serves the purpose to accredit cities for AFHC awards and might not fully enable cities to make use of the framework to collect data both qualitatively and quantitatively periodically in a systematic way for more effective evaluation and monitoring. AFHC should work with its member cities, making use of the award system to improve data collection in a more systematic way. Site visits would also be incorporated to gain a deeper understanding of the process of change and the impact. However, the three case studies have highlighted some successful stories of how the HC movement has made changes to the city well-being over time. 

## 5. Conclusions

The overarching theme for Phase V of European Healthy City (2009–2013) is health and health equity in all local policies [67]. “Health in All Policies” is based on a recognition that population health is not merely a product of health sector activities but largely determined by policies and actions beyond the health sector. The case studies have shown that the three cities have reached equivalent status. The cities taking time to set the stage in adopting principled values for good governance, community engagement, sustainability, and driving towards equity have been shown to yield results [67]. AFHC, through the Award system, has captured a huge database of healthy cities at different stages of development. It can play an important role to moblise cities with strong action in HC development, such as the three cities in this study, to support cities in strengthening their efforts to bring key stakeholders together to work for health and well-being, to harness leadership, innovation, and change, and to enhance their potential to resolve local public health challenges as in phase VI of European Healthy cities [25,35]. 

## Figures and Tables

**Table 1 ijerph-18-06617-t001:** Building a Healthy City for Healthy Living.

Healthy public policy	- The local political leaders should accord health high priority in public policy and integrate “Healthy City” development into existing systems within the City Council - The city should engage key stakeholders, professionals, and academics with expertise in Healthy City and community leaders to enable the concept of “Healthy City” to be attached to other policies that are more likely to receive political attention, such as housing and unemployment
Supportive Environment	- The city should address environmental factors, i.e., physical environment including clean water, adequate sanitation and unpolluted environment - The ecosystem of the City is stable and sustainable - Social environmentincludes mental and emotional well-being, free of crime and domestic violence; the economic environment includes housing, transport, and employment - The City is taking strong initiatives in valuing diversity, respecting and preserving the social and cultural values of the communities
Strengthening community action	- Organise local seminars/workshops for the local citizens to explore the concept of Healthy Cities and the applicability to encourage communities to participate in urban development for better health and quality of life - Creating a city health profile to understand the community perception of health, important health issues of the city, and programmes meeting the needs of the community
Advocacy	- Integration of efforts of different parties and stakeholders within and outside the health sector to play an advocacy role in incorporating health considerations into urban development and managing a clean and safe environment, including housing quality and greater use of renewable energy - High degree of participation and control by the public over the decisions affecting their lives, health, and well being

**Table 2 ijerph-18-06617-t002:** Health promotion in diverse settings and strategic plans to enhance sustainability.

Gangdong-gu	Owariasahi City	Kwai Tsing District
Health Promotion in Diverse Setting: School Setting		
Supportive services to school by City The city assigned “Needs Call Counsellor for the first time in Korea, starting with only 1 counsellor handling around 1500 cases in 2011 to 47 counsellors handling over 11,000 cases in 2017. The funding to support environmentally friendly school meals in elementary and middle schools increased from 4.3 billion Won in 2009 in 5 schools to 43.5 billion Won in 41 schools in 2018. Physical activities time increased from 40 to 80 min in school with an increasing number of schools engaged in Youth Spots climbing and Gangdong-gu Sports Leagues.	Food education at schools Dissemination and publicity of good eating habits by joint efforts between schools and families.	A special training programme by special earmarked-fund SPS Designed as “Other Learning Experience”, the new secondary school curriculum was developed to provide a first-aid training programme, including CPR together with a career talk on healthcare professions. The program has broadened their view and increased their knowledge and interest in medical and nursing professions, and nourished their positive values and attitude toward life. A school-based health program coordinated a series of appropriate developmental activities, including in-depth physical and psycho-social assessment with further follow-ups, such as medical treatment, counselling, and other adolescence health service if needed.
	Food education by the school lunch centre Provision of school lunches not containing the seven major allergens on some occasions.	
	Food education for each generation Food education seminar. Health guidance. Dental check-up.	
Park and Recreational Spaces		
The City has created urban gardens and neighbourhood spaces by expanding urban gardens from 226 lots in 2010 to 7612 lots in 2018 and park space from 6.47 m^2^ per capita to 7.3 m^2^ in 2018, and increased accessibility for physical activities by expanding bicycle roads from 34. km in 2013 to 48.55 km in 2018 and use of the roads and walking courses within the area from 36.5% to 42.6%. The area per captia living zone part has expanded from 7.09 m^2^ (2016) to 7.3 m^2^ (2018).	The Eco-garden City, as part of City’s leading plans for a Healthy City, is full of greenery.Citizens have taken part in activities to develop and maintain parks and greenery, activities to clean the environment, and cultivate curtains of green plants offering shde in summer. Events are organised in the city’s forests and reservoirs to enable citizens to enjoy the rich greenery and the natural environment.	
Villages		
The city has created energy-independent villages by developing low-energy, eco-friendly housing complex environment by promoting eco-friendly reconstruction with the number of certified complex increasing from 9 in 2013 to 14 in 2017. The city has made use of a Healthy Counselling Centre, Health Clubs, and group meetings of people with chronic illnesses to improve accessibility for NCD prevention and management, and also a Health Management Centre to provide one-stop tailored healthcare from diagnosis to prevention.	- Holding events using the forest park. - Conservation of natural forests.	As part of SPS, fall risk assessment and intervention programme with the aim to minimise injury among elderlies, especially with dementia, was conducted, covering 89% of elderly homes in the district with more than 90% of the homes showing good compliance to environmental safety with good lighting, dry floor, open hallway, and stable furniture.
Housing Complex		
The city promotes physical activities by climbing stairs in apartments with 5 housing complex participated in 2015 increased to 15 in 2017.	- Ecological ISO for homes. - Utilisation of natural energy.	Suuported by the Occupational Safety and Health Council, a set of criteria was developed as outcome measures for accreditation of a Safety and Healthy Estate aiming to strengthen the property management on key areas related to safety and health.
Safe and Healthy Workplace		
	Universal design is integrated into land readjustment projects involving roads, parks and other urban infrastructures to be maintained and updated so everyone can get around safely. Health support for citizens through cooperation between Aichi Medical University’s medical library and public libraries (ME-LI.LIN)	Survey was conducted to explore the attitudes of residents and hospital staff on safety and health issues at workplace and home jointly with Occupational Safety and Health Council with activities organized in several locations to promote safety and health.
Strategic Planning		
Gangdong-gu’s Healthy City Master Plan Goals and Strategies Korea’s first customized Healthy City Planning Guideline - The Healthy City policies have shifted from focusing mainly on health, medical care, and welfare to the empirical analysis of the relationship between urban physical environments leading to customised health city planning guideline, first in Korea. - Advisory committee members consisting of 17 people, including professors (urban planning, health, architecture), CEOs (landscape, urban planning, and architecture), and Food-Health-related research directors.	The Fifth Owariasahi City General Plan It was formulated as the general guidelines for community building. The Healthy City activities were recognized as part of the means to advance the general plan. The city reviewed the Owariasahi Healthy City Programme (HCP) that was formulated in December 2005. The projects under implementation in the city have been systemized into the three guidelines defined in the Owariasahi HCP: - Making a city that prevents people from becoming bedridden; - Making a city people want to go out into; and - Making a city where people would always want to live to promote a Healthy City with healthy people. The three guidelines had synergetic effects postponing the need for nursing care and extending healthy life expectancy. The results are fed back to citizens every year.	Signature Project Scheme (SPS) Utilised the SPS earmarked funding (HK 100 million HKD) to District Councils initiated by Hong Kong SAR Chief Executive Policy Address in 2013 to enhance Community Health Care Services in launching district-wide primary health care services. Kwai Tsing District was chosen to launch first District Health Centre in 2019 as government strategy to enhance primary health care, as announced in 2017 Policy Address.

**Table 3 ijerph-18-06617-t003:** Policy commitment and Community Partnership.

Gangdong-gu	Owariasahi City	Kwai Tsing District
The Pledge of Mayor for election (July 1 2018–June 30 2022) - Creating Healthy City environment (7 cases): creating sports facilities (4 cases), creating walkways (2 cases,) creating a park (1 case). - Creating a healthy living environment (1 case): establishing Gangdong’s standards for fine dust. - Creating a healthy environment without alienation and discrimination (1 case): expanding sports programs for people with developmental disabilities. - Health-friendly policy (1 case): establishing eco-friendly urban agriculture local food system. Establishment and revision of relevant ordinances for health city: Basic ordinance for child obesity prevention.	Owariasahi has developed and is implementing seven leading plans: The Healthy City activities initiated by the city were recognized by citizens and the city assembly, and the HC activities were recognized as a means to advance the general plan (not simply as part of measures) for the city. The HC concept was introduced in most of the city’s policies. The seven leading plans are: (1) Cycle for Good Health; (2) Enhancing Health of the Elderly; (3) City Enjoyable for Walking; (4) Getting Refreshed; (5) Food Education for Health; (6) Eco-garden City; (7) Renovation of the Whole City. They realize the three guidelines in HCP. Information about Healthy Cities was shared between the HC Promotion Headquarters (consisting of the mayor and city officials in ranks higher than director general) and the Healthy City Promotion Headquarters Promotion Committee (consisting of city officials). The Asahi Health Fiesta has been organized since 2005 to help citizens understand the concept of Healthy Cities and expanded to the concept, “Health of City in 2018”. Many events of dissemination of information of the city were held with industry–academia–administration collaboration and collaboration with agricultural organizations, in addition to previous concepts mainly focused on physical health and mental health.	First District Health Centre in Hong Kong In the 2017 Policy Address, the Government announced that it would set up a steering committee on primary healthcare development to comprehensively review the existing planning of primary healthcare services and provide healthcare services via district-based medical–social collaboration in the community. Kwai Tsing District was chosen to set up a District Health Centre (DHC) under a brand new operation mode taken reference from previous track record in Healthy City development. The Kwai Tsing DHC was inaugurated in 2019 operated by Kwai Tsing Safe City and Healthy City Association (KTSCHSA).
Health in other policies		
Housing Provide eco-friendly houses with certification mark “Eroeum” to buildings that applied a “Gangdong low-energy, eco-friendly apartment houses guideline” among apartments with over 300 households. City planning Analyzes the health level of Gangdong-gu and analyzes the relationship with the urban environmental field focusing on vulnerable areas, and presents the results. Transport Expansion of bicycle infrastructure- Create a healthy bicycle-usage environment by expanding bicycle roads and unmanned rental system. Walking-friendly walkway environment Pedestrian-priority road where the sidewalks and roads are not separated from each other. Specialized school zone safe walkways safe from traffic accidents. Specialized safe walking roads with nightlights. Heat-wave shelter and outdoor canopies repair Gangdong Green-Way and expand green spaces such as parks to increase accessibility of residents’ physical activity. Environment Preserve ecological environment - Restoring and greening project of Godeokcheon ecological stream to become a resting space in the city centre, with facilities such as waterfront stage, floor fountain, athletic park, walkway, and bicycle road. - Create Gangdong Areum Forest and restore urban forest; - Operate experiencing forest centre for children; - Duncheon-dong Ecological Landscape protected area; - Expand green spaces in living zone. Low-carbon city - Supply renewable energy, supplying eco-friendly integrated energy and building energy efficiency; - Operate eco-mileage system; - Energy-independent villages; - Create Gohdeokcheon; - Activate biodiesel with cooking oil waste. Active resource circulation - Resource circulation center; - “Pay as you throw” system in apartment houses to reduce food garbage; - Operate Gangdong flea market; Create safe city - Integrated CCTV control centre; - Scout for women’s safe way back home and safe delivery box for women.	Renovation of the Whole City Leading plans of the Healthy City Program to promote citizens’ health through the urban environment by improving the urban infrastructure, maintaining public facilities, enhancing convenience of public transport, and promoting introduction of universal design, etc. Transport “Making a city people want to go out into” (one of the Healthy City program policies), the City Public Transport Council, which is organized by general passenger automobile transportation business operators, etc., reviews the city bus service that is used by citizens. Social Welfare Social Welfare Council implements project to collect surplus food and food close to its use-by date and are still edible from general families, and distribute them to the needy and their families for free to reduce food waste. General Waste Disposal Basic Plan The City and citizens discuss and review the promotion and progress management of the plan to develop a recycling-based city by promoting 3Rs: Reduce, Reuse, Recycle. Sport and Recreation - The City collaborates with health promoters in promoting activities for muscular training, walking, laughter, and health. - Working with neighbourhood associations in park conservation, caring for small lots made into what we call pocket parks, and carrying out crime-prevention patrols and voluntary disaster-prevention activities. - Walking events planned by the Love Yada River Association, youth league of the society of commerce and industry, and sports promoters were designated as part of the Hotto Challenge Walking to encourage citizens to walk 40 km per year through walking events or on walking courses. Good Health! People–City–Partners program From 2011, the scope of the program under the HCP with companies and groups that work on creating Healthy City. A joint program was organized by the city and registered companies. The Company Cafeteria Lunch Tasting Tour to Promote Health provided citizens with opportunities to visit registered companies, learn about companies’ health-related activities, and try health-conscious meals offered by companies to their employees. Since 2013, Healthy Lunch (caloric intake: 650 kcal or less, salt: about 3.5 g) was offered every Wednesday with cooperation from a company that runs a cafeteria in the city office building. This program is intended to motivate city officials and citizens to promote health through meals.	Additionally, in the 2017 Policy Address, the Government announced that it would enhance community health through cross-sector and multi-disciplinary collaboration. - Regularise and extend the Dementia Community Support Scheme. - Based on the evaluation results of the Student Mental Health Support Pilot Scheme, consider ways to provide appropriate support services for students with mental health needs.
Community partnership		
Citizen engagement Seoul Research Institute held a large-scale citizen policy debate attended by more than 100 people, including advisory committee members, external experts, and other local government officials, where it presented research results and collected various opinions deriving high completion outcome. Connectedness with the past, the cultural and biological heritage of City dwellers Urban regeneration in 2014 with the concept of “a village where history and culture coexist with residents” in order to improve environment of area of Amsa to revitalize local communities. Examples of projects after collecting residents’ opinions, consulting with related departments and experts: - warm and prehistoric road development project; - Amsa house makeover project; - Amsa walking environment improvement project; - urban agriculture revitalization project; - Urban Regenerating Village School; - Amsa Community “Madang”, which can increase citizens’ quality of life. Gangdong Prehistoric Culture Festival Developing “exploring prehistory street parade”, where 1500 residents plan and participate and establishing storytelling marketing strategy.	Citizen engagement The City encouraged citizens to form a volunteer group, “Park Lovers Association”, and the group cleans and weeds in the parks and plazas regularly. City Development Workshop” was organised before setting up the “City Development Master Plan” reflecting residents’ opinions. Local residents established “Conference for City Redevelopment” for readjustment of Sango Station area to make the station area useful and attractive. The City built a facility “Recycle Plaza”, to dispose sorted recycle materials and to accept items for Reuse. It entrusts the management of this plaza to bodies that support the employment of elderly and disabled persons to make their lives more enjoyable.	Partnership Partnering with School of Optometry, Hong Kong Polytechnic University, and with subsidy from the Kwai Tsing District Council to provide comprehensive eye-checking and Optical Dispensing Service to Kwai Tsing residents aged 50 or above.

**Table 4 ijerph-18-06617-t004:** Information and Innovation.

Gangdong-gu	Owariasahi City	Kwai Tsing District
Creation of City Health Profile established under Healthy City Planning Guideline Health Status indicators - Life expectancy at birth, age-standardised mortality, suicide rate; - Incidence of infectious disease; - Chronic disease diagnosis management rate; - Environment disease diagnosis rate; - Subjective health level; - Stress cognition rate; - Depression experience rate; - Total fertility rate; - Infant recommended vaccination rate. Living environment - Public transportation, sports facilities, and park, food vendors distribution; - Fine dust ozone, second-hand smoke; - Traffic safety, crime, risk perception; - Health budget, medical environment satisfaction; - Traffic environment satisfaction, safety nature. Health and equity - Disease susceptibility, mental health, subjective health level; - Fruit and vegetable intake; - Unmet medical experience, vaccination, health check-up; - Subjective health level of elderly living alone, depressive experience, walking practice, falling experience. Analysis of economic and social determinants of health for the city - The analysis results of residents’ health status based on major sociological and economic determinants. - The levels of desirable health behaviours (physical activity practice, walking practice, participating in sports activity, etc.) were low in the elderly and high unhealthy behaviours (chronic/acute disease, accident experience rate, suicide ideation) were high in the elderly with low education and income. - People with low age and high education and income level have high health satisfaction through healthy behaviour practice. - Identification of major problems of the city. - Population aging and increasing prevalence of chronic disease. - Deteriorating health problems, such as increased obesity in children and youth. - Weak mental health. Innovation/new initiatives based on city health profile *Restoring Ecological Stream Godeokcheon* Godeokcheon had problems with drying, ecological cutoff, pollution, etc., but is now transformed into an ecological river where fish live and mallard ducks and raccoons come. In the planning process, specialists in various fields, such as ecology restoration, water quality, landscaping, stream construction, environmental union organisations, and public officials worked together, focusing on restoring ecology, enhancing biodiversity, improving public accessibility and utilisation. They discussed creating synergy by linking construction method, planting material, creating bridge shade shelter and Godeok commercial business complex, and formation of resident community for sustainable ecological river maintenance management. Creating an environment for physical activity - Making and distributing Walking Map in 18 apartment complexes in jurisdiction; - Health club activities once a week of 18 apartment complexes; - Install urban walking trail and Walking App in 18 places; - Install Walking Light in every 18 apartments; - Operate Employee Walking Day 8 times; - Improve physical activity rate by life cycle; - Child: operate child obesity prevention program by sending sports instructors from Gangdong Council of Sport for All; - Youth: increase physical activity class in 8 schools, 2582 people; “Moving Class”: 3 schools, 1230 people; - Adult: providing healthcare programs according to characteristics of living and working space; - Senior: open a class 2 times a week, sending physical activity leader to 25 senior citizen centres. Early diagnosis and management of dementia - Dementia check-up; - Dementia management and prevention education; - Cognition rehabilitation program for dementia patients; - Building Memory School. Mental health check-up and suicide prevention - Mental health centre and suicide prevention centre; - Early detection and treatment of depression through depression check-up; - Proper perception education on mental health and mental health experiencing class; - Enhance life-cycle high-risk management system; - Healthy 100s Counselling Centre Program. Health check-up (blood pressure, blood sugar, blood lipid) - Individual health counselling (physicians, trainer and nutritionist); - Early detection and registration of metabolic syndrome and management of continuous dosage; - Sending detected chronic disease patient to nearby private medical institutions; - Elderly health check-up and counselling: continuous health care of visiting nurse from Visiting Community Centre; - Self-help group of patients with hypertension, diabetes, hyperlipidemia; - Operate Health Clubs: 58 clubs; - Foster community health leaders: 104 people; - Health Management Center; - Integrated management of Internal Medicine and Metabolic Syndrome Management Center; - One-stop tailored health care from diagnosis to prevention; - Self-help group of patients with hypertension, diabetes, hyperlipidemia; - Mobile Health Care Using Smart Phone; - Child obesity prevention management: tailored integrated education such as Moving Schools; - Healthy School: 3 schools, physical activities, nutrition education; 64 schools, child obesity prevention programmes for children in vulnerable classes such as Community Child Centre: 6 places. *Involving private sectors in the development of healthy lifestyle for the citizens* - “Healthy 100s Counselling Center” was set up at all 18 community centres of the apartment complexes Each centre has nurse practitioner to improve lifestyles and detect metabolic syndrome and chronic illnesses early by 1-to-1 health counselling and health assessment such as abdominal circumference, blood pressure, blood sugar, triglyceride, good cholesterol, etc. It is a community-based mini health centre managed by and affiliated with local medical institutions.	Health Information The Healthy Asahi 21 Plan has been formulated for city development based on recognition that the health of individual citizens is significant and important for sustainable society. Statistic data and questionnaire result are used and evaluated by extracting current situations and issues surrounding health promotion. The current values and target values are as follows: Nutrition/dietary habits; Physical activities/exercise; Recreation/promotion of mental health; Smoking/COPD; Alcohol; Dental health; Non-communicable diseases; Health of parents and children; Food education. The Cycle for Good Health and Enhancing Health of the Elderly (leading City plan) identifies users’ health conditions through measurement, provides health promotion courses appropriate for health conditions (e.g., bringing physical checkup Results to normal conditions, easy muscular training exercise), and offers appropriate advice. Results from the Questionnaires for City Development, questionnaire regarding the Second Healthy Asahi 21 Plan, questionnaire conducted at the Asahi Health Fiesta, and questionnaire relate to Asahi Health Meister show steady and certain improvement in citizens ‟health awareness” according to the effects from various efforts of Healthy City Program. About 90% of citizens answered that they think they are healthy according to the result of questionnaires. Innovation/new initiatives based on city health profile The General Plan Deliberative Council was set up to reflect opinions of various stakeholders when formulating the comprehensive local strategy and local population vision and verifying the effect of the comprehensive local strategy. The meeting consists of public organisations (e.g., neighbourhood associations, chamber of commerce, and industry), academic experts, and citizens selected through public solicitation. The City hold the Children’s Conference, in which the city mayor and the superintendent of the board of education listen to the opinions of students and discuss how to improve the city to be comfortable and attractive. The City will reflect the opinions of students so students will gain their own opinions and interest in city development by that chance. There are three “Children’s Canteens” run by NGO and Social Welfare Service Corporation in the city. In the Children’s Canteen, many children can get together for a meal, including children who cannot eat sufficiently due to their family’s economic situation and children who often eat alone because family members cannot stay at home for children. The Children’s Canteen are open to children and people of all ages and offer meals inexpensively. Guidelines for Establishing Owariasahi Healthy City Promotion HQ: organizational regulations for the city’s executive officials to develop comprehensive measures related to HC through cross-sectional involvement. The city concluded the Agreement regarding Cooperation in Health Promotion and Emergency Response with Pharmaceutical companies. The city works with business operators to implement measures for preventing heat stroke and metabolic syndrome and set up food vending machines that offer products free of charge in the event of disasters. The Koala Class program is organised for children (about 18-month-olds to before nursery school age) who may have developmental problems (including speech) and their parents. It aims to encourage parents to play with children, ask for advice through discussion meetings among mothers, and review the parent–child relationship. The Child Care Support E-mail program is organised to disseminate information that is useful for expectant mothers and families with children to alleviate concern about childbirth and child care. Asapy Vaccination Navigation Schedule helps to manage complication due to the types and frequencies of vaccinations. The vaccination schedule can be managed on registered terminals (e.g., PCs, smartphones). Notices are sent to the terminals by e-mail. The service also enables users to check the information about vaccination and infectious diseases as well as the schedule of health check-ups and health consultation for infants.	Health Information by various academic studies Suicide and Self-Harm Conducting supportive research for evaluating interventions and gaining knowledge in preventing self-harm and suicide Injury Prevention Interventional trial of surveillance-based prevention programme for injuries in elderly homes. Violence and abuse Study of domestic violence aims to examine the association between domestic violence and clinical outcomes. Traffic Injury Database Develop an integrated traffic injury database, incorporating the Transport Department, Injury Surveillance System, and Princess Margaret Hospital Accident and Emergency to facilitate epidemiological studies with GIS analysis together with this survey. Innovation/new initiatives based on city health profile - The Kwai Tsing Safety and Healthy Charter was launched in 2006, with sectors including private ones signing the charter. The charter was signed again in 2014 to strengthen the networking and cooperation among major government departments, public utilities, education institutes, private organisations, NGOs, and hospitals in maintaining a healthy and sustainable health care system. - Services to help people with disabilities, ethnic minorities, new arrivals, and the underprivileged to integrate into the community.

**Table 5 ijerph-18-06617-t005:** Research and Resources.

Gangdong-gu	Owariasahi City	Kwai Tsing District
Framework for Needs Assessment Community Health Survey to investigate the needs of Gangdong-gu residents in health fields such as health behaviour of the residents and medical use. Social: investigate the needs of Gangdong-gu residents’ daily life and social problems in general, and online surveys on the administration and policies of Gangdong-gu. Conducting “local residents’ health priorities survey” in order to establish local health care plan. Actively collecting residents’ opinions through creating meeting place with the Major including “Residents’ Discussion Session for Vision and Regional Development in Gangdong” in order to establish the Healthy City Master Plan (2016) and the Gangdong-gu Healthy City Planning Guideline (2018). Gangdong-gu has used the indicators of the Community Health and the Gangdong Social Survey to construct health city profiles to monitor the performance of healthy city projects and programmes.	Needs assessment and evaluation - Questionnaire study to identify citizens’ views on health and their daily habits” upon preparation of the Second Healthy Asahi 21 Plan. - Questionnaire study every 2 years to assess how far targets set in the policies and key programmes in the Fifth General Plan have been achieved and for use as a basis in further city planning.	Research projects - Development of the Injury Surveillance System (ISS)”: a web-based system with ICD coding and GIS-based injury surveillance was used to integrate injury data with geospatial analysis. - Injury Prevention and Safety Promotion: projects include Traffic Injury Database, Suicide and Self Harm, Violence and Abuse, and Injury prevention in RCHE.
Evaluation index indicating the health risk factors of Gangdong-gu is changing positively Current smoking rate: 24.4% in 2009 to 17.4% in 2017; Practice rate of moderate to severe physical activity: 17% in 2009 to 28.6% in 2017; Current smoking rate: 24.4% in 2009 to 17.4% in 2017; Weight control trial rate: 54.5% in 2009 to 70.4% in 2017; Hypertension medication rate: 82.2% in 2009 to 89.4% in 2017; Diabetes medication rate: 83.9% in 2009 to 91.2% in 2017; Subjective health level perception rate: 42.2% in 2009 to 50% in 2017.	Achievements In the Questionnaire on City Development in 2018, about 53% of citizens replied that they know about the city’s efforts to become a Healthy City, almost doubled since 2006 (about 26%). The percentage of elderly citizens requiring nursing care/cost of care for elderly has remained low. The average healthy life expectancy and independent life duration have risen. The collaboration with private business operators has been strengthened. The Healthy City activities have been employed as means to advance the new General Plan of the city.	Achievements Changes between 2015–2016 report and 2016–2017 report on health behaviours: - 59% increase in health check-ups; - 35% increase in primary eye care consultation; - 18% increase in nursing consultation; - 28% in drug enquiry.
Resources Gangdong-gu is supporting Healthy City Team and project coordination. Subsidies received in 2018 were 1462 million WON (1 WON = 0.000899 USD).	Resources The City has committed resources for questionnaire study. The City has integrated the Health Festival and Asapy Smile Walking Rally into the Asahi Health Fiesta as an opportunity to publicise the City’s efforts to become a Healthy City. In 2018, the city subsidised the Health Festival Executive Committee and Asapy Smile Walking Rally Executive Committee.	Resources Ad hoc funding of 600,000 to 800,000 HKD.

**Table 6 ijerph-18-06617-t006:** Infrastructure and Intersectoral and Training.

Gangdong-gu	Owariasahi City	Kwai Tsing District
Steering Committee composed of representatives from all the sectors and local stakeholders with Deputy Mayor as Chairman and Planning Economy Director as Vice Chairman. Gangdong-gu established Gangdong-gu Healthy City Master Plan and “Healthy City Team” has been transferred to the Planning and Budget Division of Gangdong-gu Office from the Community Health Division of Gangdong-gu Community Centre in March 2017 in order to systematically promote the Gangdong-gu Healthy City Project in every field.	Round-Table Conference for a HC consists of 12 members: a professor of a local university, and representatives from senior clubs, agricultural/cooperatives, alliance of neighbourhood associations, the society of commerce and industry, social welfare council, health education teachers association, Regional Activities Communication Cooperative Group (Mothers’ Club), health-promoters association, sports-promoters association, and citizens selected through public solicitation, etc. Promotion Office for the HC, Secretary Division, Planning Dept as designated office.	Steering Committee includes doctors, professors, school principals, district councillors, NGO members, businessmen, and government officials KTSCHCA has established organisation framework for operation.
Training		
Since Gangdong-gu joined AFHC in 2008, it is sharing its major policies and projects with overseas Healthy Cities, including: - Eco-friendly urban agriculture, healthy 100s consultation centre, Gangdong Green-way; - Uploading good examples of Healthy City Policy on Gangdong-gu Website so that data can be shared anywhere; - Presented cases of Healthy City Project; - Healthy-City-related workshops and seminars; - Published and distributed Healthy-City-related reports.	A city official served as an instructor at a university, and explained the Healthy City activities to students who are expected to become future leaders of the city. The city actively accepts inspection tours and invitations to give lectures at external entities. Some members of the Promotion Office for a Healthy City participated in the seminar regarding “Sustainable Developments of Goals and Healthy City”.	Knowledge and skills transfer Over the years, many visitors from Mainland China, Macau and neighbouring countries came to Hong Kong to visit the City about our model of Safe Community.Regular meetings to discuss health-promotion planning. In line with evaluation framework by AFHC Award Committee.

## Data Availability

The data presented in this study are available on request from the corresponding author. The data are not publicly available due to privacy.

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
