# Peer review of "Engaging Diverse Community Groups to Promote Population Health through Healthy City Approach: Analysis of Successful Cases in Western Pacific Region"

_ijerph, 2021, doi:10.3390/ijerph18126617_

Round 1
Reviewer 1 Report
The article is very interesting and it would be desirable for a similar study to be repeated for other HCs in other WHO regions. The comments below are motivated by the interest aroused in the reviewer, who then asks for further elaboration or clarification of certain issues.
Since the issue of Covid-19 has had, and still has, a great impact on cities and may lead to a review of the Spirit model as well, I think this could be more critically addressed in section 1.3 and in the discussion and conclusions (section 4,5). In particular by highlighting the importance of the relationship between Social Environment, Built Environment, physical and mental health and accessibility to primary care. Also the aim (lines 136-139) should make more explicit what is meant by "complex interconnections" to integrate ecological, social, economic aspects to promulgate healthy city development. The risk is that the desire to overcome the "medical model" leads to ignoring the problem of accessibility ‘for all’ to social and health services.
In section 2 could be clarify for the reader the scope of the AFHC and how the SPIRIT framework was adopted within the AFHC. In particular, explain whether the diversity of scale (large cities, small cities, districts, etc.) has been taken into account in application of the SPIRIT model.
Section 3 is very interesting, but it is a little difficult to read due to the heterogeneity of the experiences of the 3 cities compared under the different topics. Perhaps it could be highlighted in illustrating each case and each experience ‘who does what and for whom’: the role of the public, the private sector, associations, and whether it is aimed at certain categories of citizens or not.
In the Discussion (sect. 4) it is stated that the analysis of the cases can highlight how HC would create conditions for better governance, enhanced inter-sectoral action and opportunities for Health in All Policies as well as greater resilience of programs facing resource limitation and political turmoil. The aspect of resilience could be discussed more in the examination of the three case studies, as well as the relationship between policies success and social and cultural inequalities in the population of the 3 cities/districts studied. There is only a hint on this point in the supplement to the article about K&T district.
The results (sect. 5) are well presented. It would be interesting to develop a little more the considerations about the K&T experience and the social impact of academic research and experimentation on a pilot case.
Also good is the indication of the limitations of the case studies and the future developments that AFHC can promote.

Author Response
Thank you very much for the expert comments of three reviewers on the invited paper “Engaging diverse community groups to promote population health through Healthy City approach: Analysis of successful cases in Western Pacific Region.” for the special issue “Working with Communities to Promote Health”. We have revised the paper accordingly as well as refined the writing.
We include our point-to-point response to the comments of the three reviewers.

Reviewer 2 Report
This study attempts to engage diverse community groups. The idea of an integrated and holistic approach is interesting. Below are the areas where the authors should further clarify to improve the quality of the paper
1. Abbreviations need to be spelled out the first time when they are mentioned in the text (e.g., SPIRIT in line 31).
2. Line 47:
One of the keywords is built environment.
However, the authors do not describe the meaning of the built environment for this paper.
3. Table 1, Line 161.
Local seminars/workshops are assigned as strengthening community action.
How do the authors feel confident that the workshop will be implemented not only the formal action?
4. Table 2
The parameter of the park and recreational spaces reveal that green spaces are adequate for citizens.
However, I am worried that not all citizens who live in the suburbs having access to those parks.
Could the authors provide the spatial data about the distribution of the green space?
Since I thought the green environment is one essential factor of the built environment discussed in this paper.
5. Table 3 and 4.
Could the authors simplify the table in graphs?
Thus, the readers can easily read and summarize the comparison.
Author Response

(The authors gave the same response as above.)

Reviewer 3 Report
General Comments: The purpose of this manuscript is to examine the concept of Healthy City fits into the complexity of health and how it can address socio economic and political determinants of health. Major points to consider: • Although you provide very interesting support from various areas of healthy living, the Healthy City concept needs to be at the very beginning of the manuscript before to begin discussing environment as health determinant or the built environment. • The aim of the paper needs to be articulated early in the Background section so the reader knows what you are building up to with your writing. • Why did you include section 2.1.1-2.1.6 where you did? I am not seeing the logic of your organization here…is this a part or related to the SPIRIT Framework? • You may wish to mention the use of case studies to support your argument in the very beginning of the Methods section when you introduce the SPIRIT Framework. Minor points to consider: • I would mention the Western Pacific region as the focus in your abstract as well. • The authors used the passive voice quite a bit in their writing and may consider changing this to active voice. • Minor grammar errors with incorrect noun verb agreement. • Some missing “a” and “the” in various places which make the sentences sound choppy and diminish flow of writing. I appreciate this manuscript and the attention it brings to the Healthy City concept. Thank you for your contribution to this important systems discussion.Author Response
Thank you very much for the expert comments of three reviewers on the invited paper “Engaging diverse community groups to promote population health through Healthy City approach: Analysis of successful cases in Western Pacific Region.” for the special issue “Working with Communities to Promote Health”. We have revised the paper accordingly as well as refined the writing.
We include our point-to-point response to the comments of the three reviewers.

Round 2
Reviewer 3 Report
Thank you for addressing the comments provided. I do believe the enhanced description of the Healthy Cities concept and the SPIRIT Framework have improved the flow the manuscript.
Author Response
Dear Editor,
Thank you for the comments from expert on the revised version. We have responded accordingly.
Comments and Suggestions for Authors
Thank you for addressing the comments provided. I do believe the enhanced description of the Healthy Cities concept and the SPIRIT Framework have improved the flow the manuscript.
English language and style are fine/minor spell check required
Response: We are very thankful for the support of reviewer. We have gone through the manuscrpt and have corrected minor spelling mistakes and minor style changes.
Thank you for the kindest expert advice. I attach the final manuscript with minor track changes.
Best regards,
Albert Lee
